# Heterophilic Graph Invariant Learning for Out-of-Distribution of Fraud Detection

Lingfei Ren
renlingfei@whu.edu.cn
School of Computer Science, Wuhan
University, China

Ruimin Hu*
hrm@whu.edu.cn
School of Computer Science, Wuhan
University, China

Zheng Wang
wangzwhu@whu.edu.cn
School of Computer Science, Wuhan
University, China

Yilin Xiao
yilin.xiao@connect.polyu.hk
Hong Kong Polytechnic University,
China

Dengshi Li
reallds@126.com
School of Artificial Intelligence,
Jianghan University, China

Junhang Wu
wjh920925@whu.edu.cn
College of Information Science and
Technology, Shihezi University, China

Jinzhang Hu
hujinzhang@whu.edu.cn
School of Computer Science, Wuhan
University, China

Yilong Zang
zangyl@whu.edu.cn
School of Computer Science, Wuhan
University, China

Zijun Huang
huangzijun@whu.edu.cn
School of Computer Science, Wuhan
University, China

## ABSTRACT

Graph-based fraud detection (GFD) has garnered increasing attention due to its effectiveness in identifying fraudsters within multimedia data such as online transactions, product reviews, or telephone voices. However, the prevalent in-distribution (ID) assumption significantly impedes the generalization of GFD approaches to out-of-distribution (OOD) scenarios, which is a pervasive challenge considering the dynamic nature of fraudulent activities. In this paper, we introduce the Heterophilic Graph Invariant Learning Framework (HGIF), a novel approach to bolster the OOD generalization of GFD. HGIF addresses two pivotal challenges: creating diverse virtual training environments and adapting to varying target distributions. Leveraging edge-aware augmentation, HGIF efficiently generates multiple virtual training environments characterized by generalized heterophily distributions, thereby facilitating robust generalization against fraud graphs with diverse heterophily degrees. Moreover, HGIF employs a shared dual-channel encoder with heterophilic graph contrastive learning, enabling the model to acquire stable high-pass and low-pass node representations during training. During the Test-time Training phase, the shared dual-channel encoder is flexibly fine-tuned to adapt to the test distribution through graph contrastive learning. Extensive experiments showcase HGIF's superior performance over existing methods in OOD generalization, setting a new benchmark for GFD in OOD scenarios.

*Corresponding author.

## KEYWORDS

graph-based fraud detection, out-of-distribution generalization, graph invariant learning, heterophilic graph contrastive learning

**ACM Reference Format:**
Lingfei Ren, Ruimin Hu, Zheng Wang, Yilin Xiao, Dengshi Li, Junhang Wu, Jinzhang Hu, Yilong Zang, and Zijun Huang. 2024. Heterophilic Graph Invariant Learning for Out-of-Distribution of Fraud Detection . In *Proceedings of the 32nd ACM International Conference on Multimedia (MM '24), October 28-November 1, 2024, Melbourne, VIC, Australia*. ACM, New York, NY, USA, 9 pages. https://doi.org/10.1145/3664647.3681312

## 1 INTRODUCTION

Recently, there has been a surge in interest in graph-based fraud detection (GFD) due to their exceptional efficacy in identifying fraudulent activities within multimedia data, such as online transactions, product reviews, or telephone voices. For example, identifying telecom fraudsters based on call voice, text message content, Internet records, and online purchase history has become a hot research topic [25]. Among these methods, graph neural networks (GNNs) have emerged as prominent tools [8, 13, 33, 42]. GNNs aggregate information from neighbouring nodes and iteratively update node representations, thereby enabling accurate identification of fraudulent entities [14].

However, a significant challenge faced by most GNN-based algorithms stems from their reliance on the in-distribution (ID) assumption, presupposing that the test data adheres to the same distribution as the training data[11, 20, 38]. Unfortunately, this assumption is often invalid in fraud detection scenarios, where data extracted from complex systems is often out-of-distribution (OOD). For instance, owing to the subjective annotation preferences of human experts, fraudsters exhibiting significant homophily ((i.e., those closely linked to annotated fraudsters) tend to be more likely to be annotated, leading to a structural shift between the training and test set [11]. Moreover, the inherently dynamic nature of fraudulent behaviour gives rise to attribute and structural shifts in collected fraud data across different timeframes [26]. According to

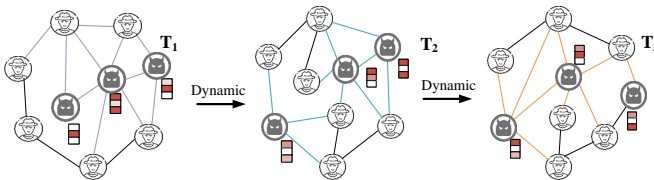

**(a) Dynamics of fraudulent behaviour lead to attribute and structural shifts**

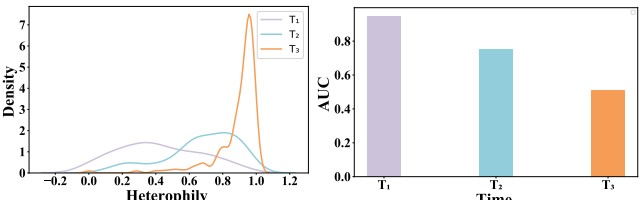

**(b) The probability density distribution of graph heterophily** **(c) AUC of trained GNN-based model at time $T_1$, $T_2$ and $T_3$**

**Figure 1: An empirical study of the OOD generalization problem for GFD approaches. Although the GNN-based model trained on fraud graphs collected at time $T_1$ efficiently detects fraudsters in the training domain, its performance degrades dramatically on the graphs collected at $T_2$ and $T_3$ due to attribute shifts and structure shifts.**

[28], GNNs are prone to exploit the shortcut feature for decision-making, which is noncausal but discriminative, leading to a marked decline in performance on test data misaligned with the training distribution. As shown in Fig 1, GNN-based algorithms trained in $T_1$ show poor generalization in $T_2$ and $T_3$.

The OOD generalization of GFD approaches is an unexplored problem for two primary reasons. Firstly, collecting fraud data poses significant challenges, often resulting in insufficient samples to train a generalized model. Second, complex distributional drifts may exist between the test and training graphs containing attribute and structural shifts. While AugAN [44] attempts to improve the generalization of fraud detection models using meta-learning algorithms, multiple observed training environments are required, and the OOD generalization based on a single training graph remains unexplored. Invariant learning has demonstrated efficacy in capturing invariant patterns, effectively disregarding spurious correlations across diverse training environments [1, 2, 4]. This prompts us to exploit invariant relationships between features and labels by constructing multiple virtual training environments with varied distributions, offering potential improvements in the OOD generalization of GFD approaches.

Developing this intuitive solution is not trivial. Firstly, there's the issue of training environment augmentation in fraud detection scenarios. EERM [37] introduced structure generators to construct diverse virtual environments, albeit at high training costs. Furthermore, while heterophily emerges as a critical factor impacting GFD approaches performance [10, 35], structure generators cannot ensure sufficiently generalized graph heterophily distributions within virtual training environments. Secondly, adapting to downstream fraud detection tasks poses another challenge. Although

existing invariant learning algorithms can extrapolate to distributions similar to the training environments, generalizing to diverse fraud detection tasks remains arduous, considering the inability to access the test set distribution beforehand. FLOOD [21] incorporated a bootstrapped learning component to understanding the test graph distribution. However, bootstrapped representation learning primarily applies to homophilic graphs, making learning the distribution of fraud nodes in heterophilic graphs challenging.

To address the above challenges, we propose a **H**eterophilic **G**raph **I**nvariant learning **F**ramework (HGIF) for OOD generalization of GFD. For the first challenge, we employ edge-aware augmentation to generate multiple virtual training environments. This approach is computationally efficient and enables the construction of diverse heterophily distributions by adjusting edges (Fraud-Fraud, Fraud-Normal and Normal-Normal edges) with different probabilities. As for the second challenge, we design a shared dual-channel encoder for learning node high-pass and low-pass representations, respectively. This encoder is trained through heterophilic graph contrastive learning and variance risk extrapolation. During the test-time training phase, the heterophilic graph contrastive learning module fine-tunes the shared two-channel encoder to fit the distribution of test graphs. In this way, the output representation of the shared encoder can generalize well to the test distribution and mitigate the effect of the distribution shift. Our contributions can be summarized as follows:

- We formally define and investigate the novel problem of OOD generalization of GFD approaches, aiming to improve the accuracy of models on unseen test graphs using a single training graph, a scenario that is more aligned with real-world application than multiple training graphs setting.
- We propose a heterophilic graph invariant learning framework, HGIF, which has an edge-aware augmentation module, a shared dual-channel encoder, and heterophilic graph contrastive learning. The encoder can be fine-tuned during the test-time training phase to fit the test distribution.
- We conduct extensive experiments to validate the effectiveness of our approach for OOD generalization. Additionally, we analyze the contributions of critical components of HGIF to its overall performance and assess sensitivity to training parameters. The dataset and code are publicly available on Github: https://github.com/Ling-Fei-Ren/HGIF.git.

## 2 RELATED WORK

**GNN-based Fraud Detection** can be defined as an unbalanced binary classification task that identifies outlier nodes that deviate significantly from the majority [19, 36]. This paper aims to perform node-level fraud detection on static graphs. The GNN-based fraud detection model mainly uses graph space-based and graph spectrum-based methods. The graph space-based methods are including GraphConsis [24], CARE-GNN [8], FRAUDER [42], PC-GNN [22], and GAGA [34]. The graph spectrum-based methods are are including H2-FDetector [27], AMNet [3], BWGNN [29], and GAD [10]. Although some encouraging progress has been achieved, these methods are based on ID assumptions, and the models have poor OOD generalization ability.

**Graph Invariant Learning** builds on the invariance principle to address graph OOD generalization. The invariance principle assumes invariance within the data so that such invariance can be found in multiple environments, thus achieving OOD generalization [16]. MoleOOD [40], CIGA [6], StableGNN [9], and GIL [17] investigate OOD generalization on graph classification. Different from them, we consider the OOD problem of node-level tasks on graphs. EERM [37], SRGNN [45] and FLOOD [21] can be applied to node-level tasks, but neither can be directly applied to GFD due to graph heterophily.

**Graph Contrastive Learning** aims to learn low-dimensional node representations on graphs without any supervised labels, like DGI [31], MVGRL [12], GraphCL [41], BGRL [30], and CCA-SSG [43]. However, due to the nature of low-pass filtering, existing graph contrastive learning methods tend to smooth the representation of each connected node pair and are, therefore, difficult to apply to heterophilic graphs. Different from them, GREET [23] and PolyGCL [? ] consider graph contrastive learning in heterophilic graphs. However, GREET designs edge discrimination to determine the edge type, leading to high training costs; PolyGCL uses different encoders between different views and cannot be combined with graph invariant learning.

## 3 DEFINITION

The paper's notation is first introduced, and then a formal definition of the problem is given. Given $\mathcal{G} = (\mathcal{V}, \mathbf{A}, \mathbf{X}, \mathbf{Y})$ denotes an attribute fraud graph, where $\mathcal{V} = \{v_1, v_2, \cdots v_N\}$ denotes node set, $\mathbf{A} \in [0, 1]^{N \times N}$ denotes adjacency matrix of graph $\mathcal{G}$ and $\mathbf{A}_{i,j} = 1$ denotes the edge between node $v_i$ and node $v_j$. $\mathbf{X} \in \mathbb{R}^{N \times d}$ denotes the feature matrix and $d$ is the dimension, and $\mathbf{Y} = \{y_1, y_2, \cdots y_N\}$ is the label matrix and $y_i = 0$ is benign node while $y_i = 1$ is fraud node.

**Out-of-distribution generalization of graph-based fraud detection.** In general, the purpose of OOD generalization of GFD approaches is to maximize the fraud detection performance on an invisible test distribution by using a limited number of training graphs and assuming that the test distribution has an OOD shift from the training distribution. Since obtaining graph fraud detection graphs is challenging, this paper aims to improve the OOD generalization of GFD approaches based on a single training graph.

Formally, Given a training graph $\mathcal{G}_{train}$ and set of test graphs $\{\mathcal{G}_{test}^i | i \in \{1, 2, \cdots U\}\}$, there are distribution shifts between training graph and test graphs: $P_{test}^i(\mathbf{X}, \mathbf{A}) \neq P_{train}(\mathbf{X}, \mathbf{A})$, and they do not share any nodes and edges: $\{\mathcal{G}_{test}^i \cap \mathcal{G}_{train} = \emptyset\}$. The goal of OOD generalization of GFD is to learn an optimal graph encoder $f_{w^*}(\cdot)$ that can effectively detect fraudsters in the testing graphs:

$$f_{w^*}\left(\mathcal{G}_{test}^i\right) \to \hat{\mathbf{Y}}_{test} \tag{1}$$

**Heterophily.** Given connections between nodes and their first-order neighbors, an edge is called a heterophilic connection if the source and target nodes of the connection have different labels (i.e., fraud node and benign node). The heterophily of node can be defined as:

$$hetero(v) = \frac{1}{|N(v)|} |u : u \in N(v), y_v \neq y_u| \tag{2}$$

where $|N(v)|$ is the number of first-order neighbours of node $v$.

## 4 METHODOLOGY

To improve the OOD generalization ability of GFD approaches, we propose a novel heterophilic graph invariant learning framework, as depicted in Fig 2.

Specifically, we first employ edge-aware augmentation to obtain multiple virtual training environments with diverse heterophily to improve the robustness of the model to heterophily distribution. Secondly, we use a dual-channel encoder to obtain stable high-pass and low-pass representations of nodes. Next, a Variance Risk Extrapolation is designed to train the encoder for OOD generalization. Thirdly, two training environments that are randomly selected are considered as different views, and a heterophilic graph contrastive learning shares the dual-channel encoder and is trained jointly with Variance Risk Extrapolation. Lastly, the shared dual-channel encoder is updated in the test-time training phase to obtain a better representation of the test set.

### 4.1 Edge-aware Augmentation

First, we construct $M$ virtual training environments with different node features and heterophily from the original training graph $\mathcal{G}_{train} = (\mathbf{X}, \mathbf{A})$. We performed two typical graph augmentation, i.e., feature augmentation [41] and structural augmentation [7].

We randomly mask the initial node features in different dimensionality for feature augmentation with a $p_s$ probability. For $k$-th training environments, we sample a binary vector $\mathbf{S}^k \in \mathbb{R}^{1 \times F}$ from the Bernoulli distribution with a probability of $\left(1 - p_s^k\right)$, i.e., $\mathbf{S}_i^k \sim Bernoulli\left(1 - p_s^k\right), i \in \{1, \cdots F\}$, and perform element-wise multiplication with the features of each node:

$$\bar{x}_i^k = x_i^k \odot \mathbf{S}_i^k \tag{3}$$

For structural augmentation, we focus on graph heterophily and hope to construct $M$ training environments with a wide range of heterophily ratios to improve the generalization ability of the fraud detection model in graph heterophily distributions. For $k$-th training environment, we sample three binary masking matrix $\mathbf{E}_{N,N}^1$, $\mathbf{E}_{N,F}^1$, $\mathbf{E}_{F,F}^1 \in \{0, 1\}^{N_i \times N_i}$ from the Bernoulli distribution with three probability of $\left(1 - p_{N,N}^{k,1}\right)$, $\left(1 - p_{N,F}^{k,1}\right)$, $\left(1 - p_{F,F}^{k,1}\right)$ for edges between normal users (N-N), edges between fraudsters and normal users (F-N), and edges between fraudsters (F-F). The masked adjacent matrix $\bar{\mathbf{A}}_1$ can be obtained:

$$\bar{\mathbf{A}}_1 = \left[ \begin{array}{cc} \mathbf{A}_{N,N}, & \mathbf{A}_{N,F} \\ \\ \mathbf{A}_{F,N}, & \mathbf{A}_{F,F} \end{array} \right] \odot \left[ \begin{array}{cc} \mathbf{E}_{N,N}^1, & \mathbf{E}_{N,F}^1 \\ \\ \mathbf{E}_{N,F}^1, & \mathbf{E}_{F,F}^1 \end{array} \right] \tag{4}$$

where the $\mathbf{A}_{N,N}$, $\mathbf{A}_{N,F}$ and $\mathbf{A}_{F,F}$ are subgraph adjacent matrix of N-N, F-N, and F-F, respectively. Similarly, we can also obtain added adjacent matrix $\bar{\mathbf{A}}_2$:

$$\bar{\mathbf{A}}_2 = \left[ \begin{array}{cc} \mathbf{A}_{N,N}, & \mathbf{A}_{N,F} \\ \\ \mathbf{A}_{F,N}, & \mathbf{A}_{F,F} \end{array} \right] \odot \left[ \begin{array}{cc} \mathbf{E}_{N,N}^2, & \mathbf{E}_{N,F}^2 \\ \\ \mathbf{E}_{N,F}^2, & \mathbf{E}_{F,F}^2 \end{array} \right] \tag{5}$$

where $\mathbf{E}_{N,N}^2$, $\mathbf{E}_{N,F}^2$, $\mathbf{E}_{F,F}^2 \in \{0, 1\}^{N_i \times N_i}$ are three binary adding matrix from the Bernoulli distribution with three probability of

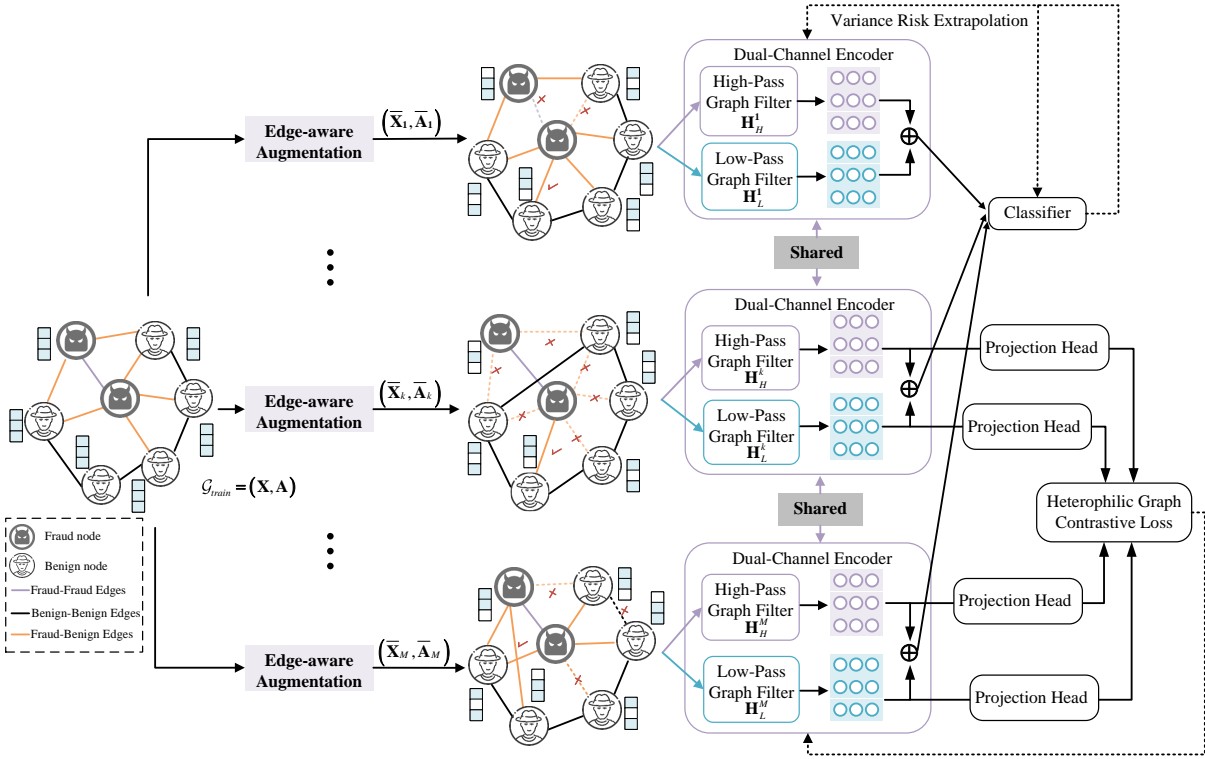

**Figure 2: An illustration of the proposed framework of HGIF. The train graph $\mathcal{G}_{train}$ is augmented by an edge-aware augmentation module to construct $M$ virtual training environments. The shared dual-channel encoder outputs stable high-pass and low-pass representations of the nodes in each environment, which are then merged and forwarded to the classifier, and a Variance Risk Extrapolation is applied to improve generalization. Meanwhile, two training environments are randomly selected to map their high-pass and low-pass representations to another latent space via a nonlinear projection head, and then the loss of heterophilic graph contrastive learning is obtained and trained with Variance Risk Extrapolation. The parameters of the shared dual-channel encoder were fine-tuned under a self-supervised task on the test set during the test-time training phase.**

$\left(1 - p_{N,N}^{k,2}\right)$, $\left(1 - p_{N,F}^{k,2}\right)$, $\left(1 - p_{F,F}^{k,2}\right)$. The the total adjacent matrix $\bar{\mathbf{A}}$:

$$\bar{\mathbf{A}} = \bar{\mathbf{A}}_1 + \bar{\mathbf{A}}_2 \tag{6}$$

Since N-N, N-F and F-F have different augmentation rates, we can obtain more training environments with varying ratios of heterophily than structure generators.

## 4.2 Dual-Channel Encoder

Recently, studies have found that high-frequency information is essential for fraud detection tasks [39]. Meanwhile, PolyGCL [?] has demonstrated that plugging learnable graph filters directly into self-supervised settings as the encoder causes performance degradation compared with the simple low-pass GCN. Thus, we propose a non-learnable filter as the encoder. Motivated by BWGNN [29], we designed a non-learnable graph filter $f_w(\cdot)$:

$$\begin{aligned}
\mathbf{H}_i^k &= W_{i,C-i}\left(MLP\left(\bar{\mathbf{X}}_k\right)\right) \\
\mathbf{H}_L^k &= MLP_L\left[\mathbf{H}_0^k \left\| \mathbf{H}_0^k \cdots \right\| \mathbf{H}_K^k\right] \\
\mathbf{H}_H^k &= MLP_H\left[\mathbf{H}_{K+1}^k \left\| \mathbf{H}_{K+2}^k \cdots \right\| \mathbf{H}_C^k\right]
\end{aligned} \tag{7}$$

where $MLP(\cdot)$, $MLP_L(\cdot)$, and $MLP_H(\cdot)$ denote multi-layer perception, $W_{p,q}$ is Beta wavelet transform defined as:

$$W_{p,q} = U\beta_{p,q}^*(\Lambda)U^T = \beta_{p,q}^*(L) = \frac{\left(\frac{L}{2}\right)^p\left(I - \frac{L}{2}\right)^q}{2B(p+1, q+1)} \tag{8}$$

where $p, q \in \mathbb{N}^+$ and $B(p+1, q+1) = p!q!/(p+q+1)!$. $\beta_{p,q}^*(w)$ is a transform of the probability density function of Beta distribution, i,e., $\beta_{p,q}^*(w) = \frac{1}{2}\beta_{p,q}\left(\frac{w}{2}\right)$ to cover the complete spectral range of Laplacian $L$ that satisfy $\lambda \in [0, 2]$.

$$\beta_{p,q}(w) = \begin{cases} \frac{(w)^p(I-w)^q}{2B(p+1, q+1)} & \text{If} \quad w \in [0, 1] \\ \\ 0 & \text{Otherwise} \end{cases} \tag{9}$$

By setting a Tunable hyperparameter $K$, we can split $W$ to $\hat{W}_L$ and $\hat{W}_H$ to capture low-pass and high-pass frequency signals, respectively. The transform $\hat{W}_{p,q}$ is:

$$\begin{aligned}
\hat{W}_L &= \left(W_{0,C}, W_{1,C-1} \cdots W_{K,C-K}\right) \\
\hat{W}_H &= \left(W_{K+1,C-K-1}, W_{K+2,C-K-2} \cdots W_{C,0}\right)
\end{aligned} \tag{10}$$

The overall representation of a node in $k$-th training environment can be obtained by connecting the low-pass and high-pass representation matrix:

$$\mathbf{H}^{(k)} = \mathbf{H}_L^k \oplus \mathbf{H}_H^k \tag{11}$$

Finally, a GNN-based classifier $f_\theta(\cdot)$ is trained to get the fraud probability. The GNN parameterized by $(w, \theta)$ is trained by minimizing the weighted cross-entropy loss defined as:

$$\mathcal{L}_{total}^k(w, \theta) = -\sum_{v_i \in \mathcal{V}} \left[ \gamma y_{v_i} \log\left(p_{v_i}^k\right) + \left(1 - y_{v_i}\right) \log\left(1 - p_{v_i}^k\right) \right]$$
$$\mathbf{P}^k = f_\theta\left(\mathbf{H}^k\right) \tag{12}$$

where $\gamma$ is the ratio of anomaly labels ($y_{v_i} = 1$) to normal labels ($y_{v_i} = 0$) in training environment.

## 4.3 Variance Risk Extrapolation

The OOD generalization is achieving low error rates on unseen test distributions. We obtain $M$ virtual training environments which are denoted as $\left\{(\bar{\mathbf{X}}_1, \bar{\mathbf{A}}_1), (\bar{\mathbf{X}}_2, \bar{\mathbf{A}}_2), \cdots (\bar{\mathbf{X}}_M, \bar{\mathbf{A}}_M)\right\}$, and according to Empirical Risk Minimization (ERM), we minimize the average loss across all training environments:

$$
\begin{aligned}
\mathcal{R}_{ERM}(w, \theta) &= E_{(\bar{\mathbf{X}}_k, \bar{\mathbf{A}}_k) \sim D}\left[ f_\theta\left(f_w\left(\bar{\mathbf{X}}_k, \bar{\mathbf{A}}_k\right), y\right)\right] \\
&= \frac{1}{M} \sum_{e=1}^M |D_e| E_{(\bar{\mathbf{X}}_k, \bar{\mathbf{A}}_k) \sim D_k}\left[ f_\theta\left(f_w\left(\bar{\mathbf{X}}_k, \bar{\mathbf{X}}_k\right), y\right)\right] \\
&= \frac{1}{M} \sum_{e=1}^M |D_e| E_{(\bar{\mathbf{X}}_k, \bar{\mathbf{A}}_k) \sim D_k}\left[ \mathcal{L}_{total}^k(w, \theta)\right]
\end{aligned} \tag{13}
$$

However, ERM is not the way to get the optimal generalization parameters since the test set may have both covariate and concept shifts compared to the training set [15]. To seek good OOD generalization, we employ the principle of Risk Extrapolation (REx) as in Eq. (14): decreasing the risk of the domain with the lowest risk while decreasing the overall similarity of the training risk to seek to increase the risk of the domain with the best performance. While this may lead to higher training risk, it also means that if the variation in distribution between training domains is amplified at the time of testing, the variation in risk will be more minor.

$$
\begin{aligned}
R_{REx}(w, \theta) &= (1 - M\lambda_{\min}) \max_e R_{ERM}(w, \theta) \\
&\quad + \lambda_{\min} \sum_{e=1}^M R_{REx}(w, \theta)
\end{aligned} \tag{14}
$$

where the hyperparameter $\lambda_{\min}$ controls how much we extrapolate. As there is a maximum value in Eq. (14), optimizing REx is difficult and unstable. To solve this problem, we replace the maximum value with the variance of the risk and obtain the V-REx to display as Eq. (15):

$$
\begin{aligned}
R_{V-REx} &= \min_{w, \theta} Var\left(\left\{\mathcal{L}_{total}^k(w, \theta) \mid k \in \{1, 2, \cdots, M\}\right\}\right) \\
&\quad + \frac{\beta}{M} \sum_{k=1}^M \mathcal{L}_{total}^k(w, \theta)
\end{aligned} \tag{15}
$$

where $\beta \in [0, \infty)$ controls the balance between the variance of risks and the mean of risks, with $\beta = 0$ leading $R_{V-REx}$ to focus entirely on making the risks equal, and $\beta \to \infty$ recovering $R_{ERM}$.

## 4.4 Heterophilic Graph Contrastive Learning

As described in the Introduction section, although the OOD generalization ability can be improved using exploratory risk minimization, it is difficult to fully adapt to the downstream tasks due to the inability to obtain the exact distribution of the test set in advance, so we design a graph contrastive learning module for learning the distribution of test set in the test-time training phase. Specifically, in the training phase, the graph contrastive learning loss participation in training with classification loss and shares the dual-channel encoder. In the testing phase, we perform graph contrastive learning on the test set to fine-tune the dual-channel encoder parameters to fit the distribution of the test set.

Although graph contrastive learning has been achieved in a wide range of studies, most of them are based on the assumption of graph homophily, which makes it challenging to apply to highly heterophilic scenarios such as fraud detection. We develop a heterophilic graph contrastive learning mechanism that extracts supervised information from high-pass and low-pass representations in different augmented views.

We randomly select two training environments as augmented views, i.e., $(\bar{\mathbf{X}}_k, \bar{\mathbf{A}}_k)$ and $(\bar{\mathbf{X}}_e, \bar{\mathbf{A}}_e)$. Before comparing the high-pass and low-pass representations, we pass the high-pass representation ($\mathbf{H}_H^k$ and $\mathbf{H}_H^e$) and low-pass representation ($\mathbf{H}_L^k$ and $\mathbf{H}_L^e$) of the two views to a nonlinear projection head $f_\varphi(\cdot)$, which maps the filtered views to another latent space. We use a two-layer perceptron network to obtain $\mathbf{Z}_H^k = f_\varphi\left(\mathbf{H}_H^k\right), \mathbf{Z}_H^e = f_\varphi\left(\mathbf{H}_H^e\right), \mathbf{Z}_L^k = f_\varphi\left(\mathbf{H}_L^k\right)$ and $\mathbf{Z}_L^e = f_\varphi\left(\mathbf{H}_L^e\right)$.

We consider the high-pass to high-pass representation and low-pass to low-pass representation of the same node between views as positive pairs and the representations of different nodes as negative pairs. In addition, to force the high-pass and low-pass filters to capture different information about the nodes, we regard the same node's high-pass representation and low-pass representation of intra-views and inter-views as negative pairs. Inspired by the InfoNCE contrastive loss [5], the graph contrastive learning loss of high-pass representation is defined as:

$$\mathcal{L}_c{}^H = -\log \frac{e^{\cos\left(z_{i,H}^k, z_{i,H}^e\right)/\tau_c}}{e^{\cos\left(z_{i,H}^k, z_{i,L}^k\right)/\tau_c} + e^{\cos\left(z_{i,H}^k, z_{i,L}^e\right)/\tau_c} + \sum\limits_{g \neq i} e^{\cos\left(z_{i,H}^k, z_{g,H}^e\right)/\tau_c}} \tag{16}$$

By combining the low-frequency representation, the total heterophilic contrasting learning loss is denoted as:

$$\mathcal{L}_c = \frac{1}{2|\mathcal{V}|} \sum_{v_i \in \mathcal{V}} \left[ \mathcal{L}_c{}^{(H)}\left(z_{i,H}^k, z_{i,H}^e\right) + \mathcal{L}_c{}^{(L)}\left(z_{i,L}^k, z_{i,L}^e\right) \right] \tag{17}$$

where $\tau_c \in (0, 1]$ is a temperature hyper-parameter for contrastive learning and $\cos(\cdot)$ is cosine similarity. In practice, we calculate the contrastive loss in a mini-batch manner [5] for large graph datasets. In the training phase, both invariant learning and graph contrastive learning are jointly optimized under the overall loss as:

$$\min_{w, \theta, \varphi} \mathcal{L}_{train} = \mathcal{R}_{V-REx} + \alpha \mathcal{L}_c \tag{18}$$

where $\alpha$ is a balance parameter. In the test-time training phase, given a test graph $\mathcal{G}_{test}(\mathbf{X}, \mathbf{A})$, we random shuffle the graph and obtain two augmented environment $\bar{\mathcal{G}}_{test}^1$ and $\bar{\mathcal{G}}_{test}^2$, the shared

**Algorithm 1:** Training process of HGIF

**Data:** Input data: $X$, Training graph: $\mathcal{G}_{train} = (\mathbf{X}, \mathbf{A})$,
Training epochs: $N_{epoch}$, Training environments: $M$.

**Result:** The prediction of nodes in $\mathcal{G}_{test}$ $(\mathbf{X}, \mathbf{A})$.

1 Initialize parameters $w, \theta, \varphi$;
2 Construct $M$ training environments via Edge-aware Augment;
3 **for** $iter \in 0, 1, \ldots, N_{epoch}$ **do**
4     **for** $k \in 1, \cdots, M$ **do**
5         Generate low-pass and high-pass representation $\mathbf{H}_L^k$ and $\mathbf{H}_H^k$ via Eq.7;
6         The representation $\mathbf{H}^k$ can be obtain via Eq.11;
7         The loss $\mathcal{L}_{total}^k$ can be obtain via Eq.12;
8     Calculate the REx $\mathcal{R}_{V-REx}$ via Eq.15;
9     Get $\mathbf{Z}_L^k, \mathbf{Z}_H^k, \mathbf{Z}_L^e$ and $\mathbf{Z}_H^e$ from $k$-th and $e$-th training environment;
10     Calculate contrastive loss $\mathcal{L}_c$ via Eq.16;
11     Train $w, \theta, \varphi$ via minimizing Eq. 18;
12 Construct two test environments $\bar{\mathcal{G}}_{test}^1$ and $\bar{\mathcal{G}}_{test}^2$ via random shuffle $\mathcal{G}_{test}$ $(\mathbf{X}, \mathbf{A})$;
13 Fine-tune $w$ and $\varphi$ via minimizing Eq. 19;
14 Return node label in the test graph $\mathcal{G}_{test}$ $(\mathbf{X}, \mathbf{A})$ via Eq. 20;

dual-channel encoder and the nonlinear projections are fine-tuned by minimizing the graph contrastive learning loss as follows:

$$\min_{w,\varphi} \mathcal{L}_{test} = \mathcal{L}_c\left(w, \varphi\right)\Big|\left(\bar{\mathcal{G}}_{test}^1, \bar{\mathcal{G}}_{test}^2\right) \tag{19}$$

Then, the node label in the test graph based on the fine-tuned model by the test-time training is predicted via the GNN model $f\left(\cdot, \cdot, w^*, \theta^*\right)$ as:

$$\hat{y} = f\left(\mathcal{G}_{test}, w^*, \theta^*\right) \tag{20}$$

The overall training algorithm is summarized in Algorithm 1.

## 5 EXPERIMENTS

In this section, we empirically evaluate real-world fraud detection data to answer the following research questions:

**RQ1:** Does Edge-aware Augmentation achieve training environments with more generalized heterophily than structure generator?

**RQ2:** Does HGIF outperform the state-of-the-art methods for OOD generalizations of GFD?

**RQ3:** How do the key components benefit the method performance?

**RQ4:** What is the performance concerning different training parameters?

### 5.1 Dataset

We conducted experiments on four real-world datasets targeted at fraud detection. They are YelpChi, Amazon, and two recently released transactional datasets, T-Finance and T-Social. The YelpChi dataset consists of filtered and recommended hotel and restaurant reviews from Yelp. The Amazon dataset includes product reviews in the category of musical instruments. The T-Finance and T-Social

**Table 1: Statistics of Datasets for OOD generalization of GFD.**

| Datasets | YelpChi | Amazon | T-Finance | T-Social |
|---|---|---|---|---|
| **Original Nodes** | 45,954 | 11,944 | 39,357 | 5,781,065 |
| **Original Edges** | 3,846,979 | 4,398,392 | 21,222,543 | 73,105,508 |
| **Original Feature** | 32 | 25 | 10 | 10 |
| **Environments** | 5 | 5 | 5 | 5 |
| **Nodes (avg.)** | 9,190 | 2,388 | 7,871 | 1,156,213 |
| **Edeges (avg.)** | 769,395 | 879,678 | 4,244,508 | 14,621,101 |
| **New Feature** | 52 | 45 | 30 | 30 |
| **Heterophily Ratio (avg.)** | [0.26,0.49 0.63,0.72 0.90] | [0.72,0.84 0.93,0.94 0.99] | [0.55,0.74 0.82, 0.87 0.95] | [0.01,0.05 0.10, 0.13 0.39] |
| **Anomaly(avg.)** | 14.53% | 8.97% | 4.58% | 3.01% |

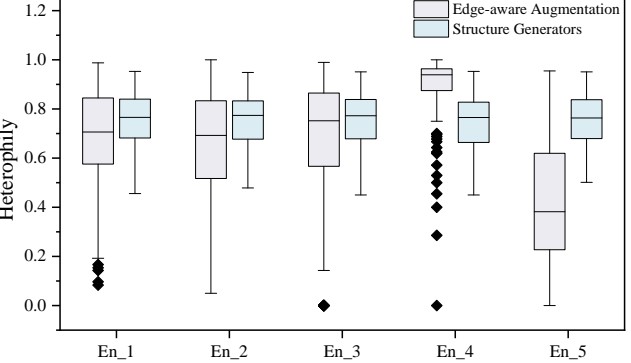

**Figure 3: An illustration of the heterophily distribution for five virtual training environments in Amazon generated by Edge-aware Augmentation and Structure Generators.**

datasets detect anomalous accounts in transactional and social networks. To construct the OOD scenarios, motivated by [44], for each dataset, 1) we randomly divide the big graph into five non-overlapping subgraphs of similar size, each subgraph is viewed as data collected in different environments; 2) generate spurious features based on node features and environment id, respectively; 3) change the graph heterophily ratios for different environments by controlling the probability of edges between the fraudsters (probability in [0.05, 0.015, 0.0075, 0.005 and 0.001]), and use the first environment graph as the training set and the rest as test sets. Table 1 summarizes the original and processed datasets' statistics.

### 5.2 Compared Methods

To demonstrate the superior performance of our proposed HGIF in OOD generalizations of GFD, we compare it with several state-of-the-art methods. The first group considers some traditional GNN-based fraud detection methods: CARE-GNN [8], FRAUDRE [42], PC-GNN [22], BWGNN [29]. The second group considers graph OOD methods: GDN [11], SRGNN [45], EERM [37], FLOOD [21].

We also compare three variants to analyze the contribution of critical modules to performance. They are HGIF\Eaa, which replaces edge-aware augmentation with structure generators. HGIF\Inv

**Table 2: Performance of fraud detection under OOD scenarios.**

| Method | Datasets | YelpChi | | Amazon | | T-Finance | | T-Social | |
|---|---|---|---|---|---|---|---|---|---|
| | Metrics | F1-macro | AUC | F1-macro | AUC | F1-macro | AUC | F1-macro | AUC |
| Traditional Methods | CARE-GNN | 45.75 | 56.28 | 53.29 | 58.28 | 50.12 | 67.28 | 45.29 | 65.28 |
| | FRAUDRE | 42.23 | 58.27 | 51.00 | 55.28 | 53.29 | 70.09 | 41.98 | 69.02 |
| | PC-GNN | 50.09 | 60.89 | 47.29 | 60.76 | 49.92 | 75.89 | 45.29 | 72.99 |
| | BWGNN | 49.01 | 54.66 | 51.04 | 59.39 | 53.25 | 74.96 | 48.03 | 74.07 |
| Graph OOD | GDN | 52.52 | 63.01 | 55.39 | 73.23 | 51.23 | 74.12 | 69.13 | 80.00 |
| | SRGNN | 53.87 | 65.12 | 61.09 | 76.23 | 50.98 | 73.16 | 73.90 | 79.72 |
| | EERM | 58.25 | 68.90 | 69.62 | 82.42 | 59.17 | 76.03 | OOM | OOM |
| | FLOOD | 52.96 | 65.01 | 67.88 | 77.90 | **59.20** | 71.79 | 79.87 | 85.01 |
| Ablation | HGIF\Eaa | 62.98 | 71.01 | 71.92 | 85.27 | 55.42 | 80.01 | OOM | OOM |
| | HGIF\Inv | 52.67 | 62.38 | 50.23 | 61.09 | 53.99 | 76.01 | 50.23 | 79.84 |
| | HGIF\TtT | 64.23 | 73.76 | 70.23 | 87.82 | 52.76 | 79.37 | 78.33 | 89.12 |
| Ours | HGIF | **66.90** | **76.49** | **76.49** | **89.75** | 57.23 | **82.02** | **82.90** | **90.02** |

removes invariant learning and performs test-time training on GNNs trained on the original graph. HGIF\TtT removes test-time training and performs prediction using an invariant model.

## 5.3 Metrics

As graph-based fraud detection is a class-imbalanced classification problem, this paper utilizes two widely adopted metrics: F1-macro and AUC. F1-macro considers the weighted average of F1 scores across multiple classes, and AUC is the area under the ROC Curve. Specifically, we show the average F1-macro and AUC of the four test sets of 5 runs.

## 5.4 Hyper-parameter Settings

Our method selects the Adam optimizer with a learning rate $\text{lr}_{train} = 1e - 3$ in the training phase and $\text{lr}_{test} = 5e - 5$ in the test-time training phase. The embedding size is set to 16, $K$ equals half the length of $W$, the training epochs $N_{epoch}$ is set to 50, the number of training environments $M$ is set to 5, $C$ in the dual-channel encoder is set to 2 for Yelchi and Amazon, 3 for T-Finance and T-Social, the training batch size $N_{batch}$ is set to 1024. $\beta$, $\tau_c$ and $\alpha$ are set to 1.0, 0.5, and 0.05, respectively. We use DGL for our algorithm implementation, and all experiments were conducted by Pytorch 1.9.0 with Python 3.8 on Ubuntu 20.04.1.

## 5.5 Heterophily Evidence (RQ1)

To answer RQ1, we investigate whether the virtual training environments generated by edge-aware augmentation have a more generalized heterophily distribution than structure generators. Specifically, we count the heterophily of fraud nodes in five virtual training environments based on the Amazon dataset, and similar results can be observed in other datasets. As shown in Fig 3, the heterophily of the training environments generated by the edge-aware augmentation has a significant variance while the heterophily of training environments generated by the structure generators is very similar. This shows that edge-aware augmentation can generate more generalized heterophilic training environments.

## 5.6 Performance Comparison (RQ2)

To answer RQ2, we evaluate the fraud detection performance of HGIF and the baseline methods in OOD scenarios. we report the results under average AUC and average F1-macro in Table 2. Accordingly, we have the following observations.

(1) **Importance of OOD generalization.** The performance of traditional GFD methods (e.g., PC-GNN, BWGNN) generally suffers a severe degradation in OOD scenarios, i.e., their generalization is poor. This is because they are all designed based on the ID assumption, and the GNN encoder tends to learn shortcut features, which can easily change in OOD scenarios, thus making the GNN classifier less generalizable.

(2) **Importance of graph-invariant learning.** The graph OOD method works better than traditional methods, proving that learning invariant features can promote the model's generalization ability. EERM performs second only to HGIF on three datasets (Yelpchi, Amazon, and T-finance), which proves that graph invariant learning can enhance the generalization ability of GNNs. In addition, the performance of EERM and FLOOD is better than that of GDN and SRGNN, indicating that the graph-invariant learning methods are more suitable for OOD scenarios than the regularization-based methods.

(3) **Superiority of HGIF.** HGIF performs best on all four datasets compared to traditional GFD and graph OOD methods. The improvements range from 3.7% to 14.8% in F1-macro and 5.8% to 11.0% in AUC. On the one hand, compared to EERM, the edge-aware augmentation used by HGIF not only avoids a large amount of training resources but also obtains diverse heterophilic training environments, which is crucial for improving the generalization of GFD. On the other hand, compared to FLOOD, the dual-channel encoder and the heterophilic graph contrastive learning designed by HGIF can efficiently learn the proper distribution of the test graph during the test-time training phase, whereas the bootstrap representation learning in FLOOD may learn the wrong distribution.

(4) **Limitation of HGIF.** We find that the F1-macro metric of FLOOD is slightly higher than that of HGIF on the T-Finance dataset, which is a very peculiar phenomenon. We find that T-Finance is the dataset with the largest average node degree (539) of the four ones (YelpChi:83, Amazon:368, T-social:12 ). HGIF uses a graph filter to

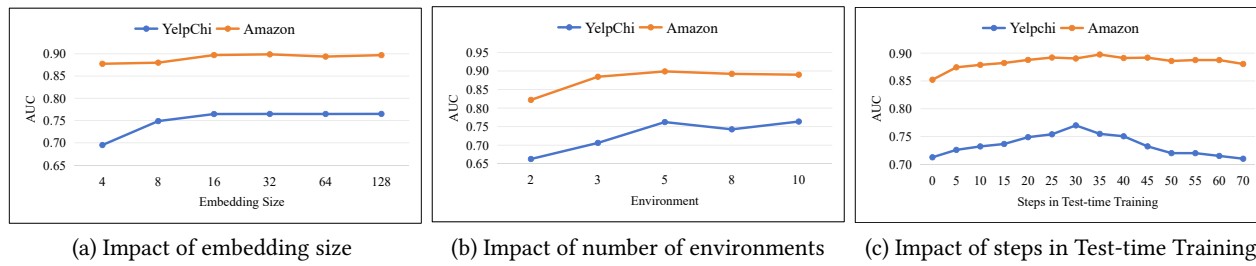

(a) Impact of embedding size  (b) Impact of number of environments  (c) Impact of steps in Test-time Training

**Figure 4: Performance of HGIF with embedding size, number of environments, and steps in the Test-time Training phase.**

adapt to the global heterophily of the graph. FLOOD adopts the GAT-like aggregation strategy, which is more sensitive to heterophilic connections in the face of large node degrees.

### 5.7 Ablation Study (RQ3)

To answer RQ3, we evaluated the key modules in HGIF by removing or replacing each module separately, i.e., edge awareness augmentation, invariant representation learning, and heterophilic graph contrastive learning. We report the results in Table 2, where the whole model HGIF consistently achieves the best scores compared to the three variants, HGIF\Eaa, HGIF\Inv, and HGIF\TtT, suggesting that each of the modules is necessary for OOD generalization of GFD.

Replacing edge-aware augmentation with structure generators results in a weaker performance than HGIF, which demonstrates the effectiveness of edge-aware augmentation in improving the OOD generalization ability of GFD. Edge-aware augmentation produces training environments with a broader range of heterophily by applying different probabilities to different types of edges (F-F and F-N and N-N), whereas the structure generators can only change the network structure. In addition, the inability of HGIF\Eaa to be applied to large-scale datasets such as T-social suggests that edge-aware augmentation reduces training resources, making large-scale graph applications possible.

After replacing heterophilic graph contrastive learning with bootstrap representation learning, HGIF\TtT has performance degradation compared to HGIF on Yelpchi, Amazon, and T-finance and a slight decrease on T-social. As shown in Table 1, Yelpchi, Amazon, and T-finance are heterophilic graphs, while T-social is a homophilic graph, and bootstrap representation learning performs well on homophilic graphs.

In the absence of an invariant learning component, HGIF\Inv is only able to fine-tune the GNN encoder through graph contrastive learning during the test-time training phase, and the results show an even more significant performance degradation than HGIF\TtT. The results concluded that combining test-time training with invariant learning is more helpful in improving the generalization.

### 5.8 Sensitivity Analysis (RQ4)

To answer RQ4, we further evaluate the sensitivity of HGIF concerning the embedding size, the number of virtual training environments, and the steps in the test-time training phase. We only report results for Yelpchi and Amazon for visual presentation. The other datasets also show similar sensitivity trends.

First, we vary the the embedding size $d$ in the range [4,128]. The result is shown in Fig 4(a). It first improves as the embedding size increases and becomes stable after 16. Considering that larger embedding dimensions require higher computational complexity, we finally set $d$ to 16 to balance performance and complexity.

Second, we vary the number of virtual training environments in the range [2,10], and the results are shown in Fig 4(b). It first improves as the number of virtual training environments increases and becomes stable after 5. This is because a smaller number of training environments may be unable to cover the generalized distribution, and training environments that are too large do not provide additional benefits. Considering that more virtual training environments require higher computational resources, we finally set the training environments to 5 to balance the performance and computational resources.

The number of steps in the test-time training phase determines how much we update the shared dual-channel encoder to fit the test distribution. As shown in Fig 4(c), it reaches best at 35 for YelpChi and 30 for Amazon. If the number of steps exceeds 35, the supervised information in the training data is lost by over-updating, even though the shared encoder may fit the target distribution.

## 6 CONCLUSION AND FUTURE WORK

In this paper, we introduce HGIF, a heterophilic graph invariant learning framework for improving the out-of-distribution generalization of graph-based fraud detection approaches. In this framework, we construct virtual training environments with generalized heterophily and use Variance Risk Extrapolation to train a shared dual-channel encoder to capture invariant features of nodes in different virtual training environments for OOD generalization. In addition, heterophilic graph contrastive learning is designed to allow the model to undergo self-supervised learning during the test-time training phase to adapt to the test distribution. Extensive experiments show that our approach improves OOD generalization for graph-based fraud detection compared to state-of-the-art GNN-based algorithms and graph OOD methods. For future work, considering that large language models (LLMs) show good generalization ability in natural language understanding [18, 32], combining LLMs with GNN may be a promising direction to improve the generalization of OOD for graph-based fraud detection

## ACKNOWLEDGMENTS

This work was supported by the National Nature Science Foundation of China (No. U22A2035, U1803262, U1736206).

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
