# OpenReview forum: "Heterophilic Graph Invariant Learning for Out-of-Distribution of Fraud Detection"
_acmmm.org/ACMMM/2024/Conference — MM2024 Poster_

### Official Review · Reviewer_vVTH · 2024-05-24

**Rating:** 2
**Confidence:** 3

**Summary:**

The paper presents the Heterophilic Graph Invariant Learning Framework (HGIF), a promising approach designed to enhance the out-of-distribution (OOD) generalization capabilities of graph-based fraud detection (GFD). The authors have effectively highlighted the framework's ability to tackle two significant challenges: the creation of diverse virtual training environments and adaptation to different target distributions.

**Strengths:**

1. The introduction to the paper is engaging and clearly outlines the motivation and objectives of the research.
2. The authors have identified and addressed two challenges: the scarcity of samples for generalized model training and the OOD generalization problem.
3. Extensive experiments showcase HGIF’s superior performance over existing methods

**Limitations:**

A primary concern revolves around the use of edge augmentation for generating both the training and testing environments. This raises the question of whether the testing environment can still be considered OOD if it is derived using the similar augmentation techniques. This approach could potentially limit the framework's ability to generalize to truly unseen distributions.

*"..Regarding the first challenge, we employ edge aware augmentation to generate multiple virtual training environments..."*

*"To construct the OOD scenarios, ... change the graph heterophily ratios for different environments by controlling the probability of edges between the fraudsters"*


Additional Comments:
1. The paper could benefit from a clearer problem statement, particularly in how it differentiates between the training and test environments in the context of OOD generalization. Specifically, it should address the issue raised in lines 235-250.  It seems that the training graph and test graph share the same graph with the same nodes and edges.
-*“ given a training graph G𝑡𝑟𝑎𝑖𝑛 = (𝑋,𝐴) …, the goal of OOD generalization of GFD is to …𝑃𝑖-𝑡𝑒𝑠𝑡 (𝑋,𝐴)|𝑖∈{1,2,···𝑈} }”.*
2.  While the paper touches on the issue of over-smoothing for certain datasets (Section 5.6), a more thorough discussion on the limitations and potential failure modes would be beneficial.
3. The presentation of Equation 16 could be improved for readability and comprehension.

**Suitability:**

2

---

### Official Review · Reviewer_mwUT · 2024-05-25

**Rating:** 4
**Confidence:** 3

**Summary:**

This paper introduces HGIF to improve the out-of-distribution generalization of graph-based fraud detection methods. The authors construct virtual training environments with generalized heterophily and use Variance Risk Extrapolation to train a shared dual-channel encoder to capture invariant features of nodes in different virtual training environments for OOD generalization. In addition, the authors design heterophilic graph contrastive learning to allow the model to undergo self-supervised learning during the test-time training phase to adapt to the test distribution.

**Strengths:**

1) This paper is technically sound, the authors propose an invariant learning framework aimed at bolstering the OOD generalization of GFD to improve the generalization of GFD, and the HGIF consists of an edge-aware augmentation module, a shared dual-channel encoder, and heterophilic graph contrastive learning.
2) The experiments are powerful and the results are convincing, the authors conduct experiments on four real-world datasets targeted at fraud detection scenarios to validate the effectiveness of the proposed HGIF for OOD generalization.

**Limitations:**

1) The motivation of this paper is not clearly described, this paper mainly solves the OOD generalization of GFD aimed at the dynamic nature of fraudulent activities, but Figure 1 in the introduction section does not match the motivation of the paper and explain in detail.
2) There are some writing problems in the paper, for example, Equation 12.

**Suitability:**

2

---

### Official Review · Reviewer_gp3y · 2024-05-25

**Rating:** 6
**Confidence:** 3

**Summary:**

In this paper, the authors introduce the Heterophilic Graph Invariant Learning Framework (HGIF), a novel approach designed to enhance the out-of-distribution (OOD) generalization of graph-based fraud detection (GFD). HGIF tackles two key challenges: generating diverse virtual training environments and adapting to varying target distributions. By utilizing edge-aware augmentation, HGIF effectively creates multiple virtual training environments with generalized heterophily distributions, promoting robust generalization against fraud graphs with varying degrees of heterophily. Additionally, HGIF employs a shared dual-channel encoder with heterophilic graph contrastive learning, enabling the model to learn stable high-pass and low-pass node representations during training. In the Test-time Training phase, the shared dual-channel encoder is flexibly fine-tuned to adapt to the test data distribution through graph contrastive learning.

**Strengths:**

1- The proposed Edge-aware Augmentation considers graph heterophily property.
2- The authors design a novel non-learnable high-pass and low-pass filter function for balancing low and high-frequency information.
3- The authors design an effective heterophilic graph contrastive learning module.
4- The experiments are comprehensive.

**Limitations:**

1-Section 6 lacks “Future Work”.
2- I encourage authors to open source code and add more experimental details.

**Suitability:**

3

---

### Official Review · Reviewer_FKdR · 2024-06-02

**Rating:** 6
**Confidence:** 3

**Summary:**

The paper introduces the heterophilic graph invariant learning framework to address the challenge of out-of-distribution generalization in graph-based fraud detection. The framework leverages edge-aware augmentation to generate diverse virtual training environments and employs a shared dual-channel encoder with heterophilic graph contrastive learning. Experiments demonstrate the superior performance of the proposed method over existing methods in out-of-distribution scenarios.

**Strengths:**

The proposed method HGIF seems novel, which combines edge-aware augmentation with a dual-channel encoder and heterophilic graph contrastive learning. The edge-aware augmentation method is well-explained and provides a computationally efficient way to create diverse training environments, enhancing the model’s ability to generalize. The use of a shared dual-channel encoder to capture high-pass and low-pass node representations allows the model to adapt to varying distributions effectively.

The paper is well-structured and easy to follow.

**Limitations:**

Although the paper compares HGIF with existing methods, it would be beneficial to include more diverse baselines, especially those that are specifically designed for OOD generalization in GFD. A more detailed discussion on why HGIF outperforms these methods would provide valuable insights.

**Suitability:**

3

---

### Meta-Review · Area_Chair_gY2G · 2024-07-03

**Recommendation:** Accept (Poster)
**Confidence:** 4

**Metareview:**

The paper has been reviewed by four expert reviewers and received mixed ratings. The major criticism concerns whether the paper truly addresses an out-of-distribution (OOD) setting. After a discussion with the SACs, the AC agrees that there is currently sufficient evidence to support the paper's OOD setting and thus recommends acceptance. The authors should carefully review the comments provided and compare the heterophily ratio differences between the training and test sets to clarify the OOD setting further.